# Urinary Potassium and Kidney Function Decline in the Population—Observational Study

**DOI:** 10.3390/nu13082747

**Published:** 2021-08-10

**Authors:** Massimo Cirillo, Giancarlo Bilancio, Pierpaolo Cavallo, Raffaele Palladino, Enrico Zulli, Rachele Villa, Rosangela Veneziano, Martino Laurenzi

**Affiliations:** 1Department of Public Health, University of Naples “Federico II”, 80131 Naples, Italy; raffaele.palladino@unina.it (R.P.); enrico.zulli@gmail.com (E.Z.); 2Department “Scuola Medica Salernitana”, University of Salerno (SA), 84084 Fisciano, Italy; giancarlo.bilancio@gmail.com (G.B.); rachelevilla@outlook.it (R.V.); rosangelaveneziano@gmail.com (R.V.); 3Department of Physics, University of Salerno (SA), 84084 Fisciano, Italy; pcavallo@unisa.it; 4Centro Studi Epidemiologici di Gubbio (PG), 06024 Perugia, Italy; mlaurenzi@comcast.net

**Keywords:** potassium, eGFR, epidemiology

## Abstract

*Background*—Some data suggest favorable effects of a high potassium intake on kidney function. The present population-based study investigated cross-sectional and longitudinal relations of urinary potassium with kidney function. *Methods*—Study cohort included 2027 Gubbio Study examinees (56.9% women) with age ≥ 18 years at exam-1 and with complete data on selected variables at exam-1 (1983–1985), exam-2 (1989–1992), and exam-3 (2001–2007). Urinary potassium as urinary potassium/creatinine ratio was measured in daytime spot samples at exam-1 and in overnight timed collections at exam-2. Estimated glomerular filtration rate (eGFR) was measured at all exams. Covariates in analyses included demographics, anthropometry, blood pressure, drug treatments, diabetes, smoking, alcohol intake, and urinary markers of dietary sodium and protein. *Results*—In multivariable regression, urinary potassium/creatinine ratio cross-sectionally related to eGFR neither at exam-1 (standardized coefficient and 95%CI = 0.020 and −0.059/0.019) nor at exam-2 (0.024 and −0.013/0.056). Exam-1 urinary potassium/creatinine ratio related to eGFR change from exam-1 to exam-2 (0.051 and 0.018/0.084). Exam-2 urinary potassium/creatinine ratio related to eGFR change from exam-2 to exam-3 (0.048 and 0.005/0.091). Mean of urinary potassium/creatinine ratio at exam-1 and exam-2 related to eGFR change from exam-1 to exam-3 (0.056 and 0.027/0.087) and to incidence of eGFR < 60 mL/min per 1.73 m^2^ from exam-1 to exam-3 (odds ratio and 95%CI = 0.78 and 0.61/0.98). *Conclusion*—In the population, urinary potassium did not relate cross-sectionally to eGFR but related to eGFR decline over time. Data support the existence of favorable effects of potassium intake on ageing-associated decline in kidney function.

## 1. Introduction

Low kidney function is a highly prevalent disorder in the population and implies the risk of kidney failure or premature death [1]. The control of hypertension is considered pivotal to reduce glomerular dysfunction and to slow down kidney disease progression although there is not unanimity about blood pressure targets and preferred antihypertensive drugs [2]. Inhibitors or blockers of the renin-angiotensin system are generally considered preferable due to specific effects on glomerular hemodynamics [2]. Similar favorable effects on glomerular dysfunction are ascribed also to antidiabetic drugs inhibiting the sodium-glucose transporter-2 [3] and to reduced protein intake [4].

The relation in the general population between dietary potassium and glomerular filtration rate is an unanswered question. Although daily potassium intake is close to the whole extracellular potassium pool, significant hyperkalemia does not occur after potassium ingestion because absorbed potassium is rapidly translocated into the intracellular space or excreted by the kidney [5,6]. The intracellular potassium translocation is due to the activity of the sodium-potassium-ATPase and/or of potassium channels. The urinary potassium excretion is activated not only via stimulation of aldosterone secretion but also due to a cascade of events in the distal convoluted kidney tubule which include potassium channel stimulation in the basolateral cell membrane, down-regulation of sodium-chloride cotransporter, and increased potassium excretion due to increased distal sodium delivery. The strict relation of potassium intake with urinary potassium excretion is the rationale which supports the assessment of urinary potassium as index of dietary potassium intake. With the use of urine potassium as objective index of dietary potassium intake, a higher 24-h urinary potassium did not independently relate to 10-year development of estimated glomerular filtration rate (eGFR) < 60 mL/min per 1.73 m^2^ in analyses controlled for baseline eGFR on 5315 adult residents in Groningen, the Netherlands [7]. Similarly, a higher spot urine potassium did not relate to 5-year eGFR decline in a population sample of 4141 adult residents in Lauzanne, Switzerland [8]. Results for urine potassium per se were not included in a study that investigated the relation of urine sodium/potassium ratio to 5-year decline of kidney function in 7063 adults from the Japanese general population [9]. With the use of food frequency questionnaires, a higher potassium intake related to a less frequent 14-year mortality rate due to kidney disease or dialysis in 544,635 retired community-dwelling US individuals [10] but did not relate to the 6-year incidence of eGFR < 60 mL/min per 1.73 m^2^ in 1780 Iranian adults [11]. Inconsistencies exist also among clinical studies as recently reviewed [12]. On the one hand, a better prognosis for kidney function was predicted by a higher potassium intake with food frequency questionnaire in Korean patients with chronic kidney disease [13] or by a higher urinary potassium in Japanese diabetics [14,15], in Dutch outpatients [16], in Korean patients with chronic kidney disease [17], and in post hoc analyses of multi-center controlled clinical trials [18]. On the other hand, a higher urinary potassium did not relate to kidney failure in a post hoc analysis of the Modification of Diet in Renal Disease cohort [19] and related positively to kidney failure or eGFR halving in a prospective study on 3939 US patients with CKD defined as eGFR < 70 mL/min per 1.73 m^2^ [20].

The Gubbio study is an ongoing, observational, longitudinal, population-based study investigating also on the relation of dietary factors with kidney diseases assessed as eGFR [21,22,23,24,25,26,27,28,29]. Measurements of glomerular filtration rate are inadvisable in population-based studies due to invasivity, costs, complexity, and lack of international standardization. The use of eGFR has progressively spread in epidemiological and clinical studies after the development of international standards for creatinine assay [30]. In particular, this was true for the CKD-Epi equation, which consistently had good accuracy and low bias over the whole range of kidney function [31]. Within the Gubbio study cohort, the relation of dietary indices with kidney function differed between cross-sectional analyses and longitudinal analyses [25,28,29]. Therefore, the present analysis was designed to investigate cross-sectionally and longitudinally the relation of urinary potassium as index of dietary potassium with eGFR and eGFR changes in adult examinees of the Gubbio study.

## 2. Materials and Methods

The Gubbio study is a population-based project ongoing in Gubbio, Italy [21,22]. The study adheres to the Declaration of Helsinki and included an informed consent and the approval by the institutional committee (CEAS-Umbria #2850/16). Study design, involvement of the invited population, response rates, and characteristics of the Gubbio cohort have been reported previously [15,16]. Three main exams were performed: exam-1 (1983–1985), exam-2 (1989–1992), and exam-3 (2001–2007). Study protocol of all exams comprised the administration of standardized questionnaires by trained physicians and a medical visit with the measurements of anthropometry and blood pressure [21,22]. With regard to collection of biological samples, the exam-1 protocol included a daytime, untimed, spot urine sample and a venous blood sample after a fast of at least two hours [21,22]. The exam-2 protocol included a timed urine collection under fed condition from the first void after the evening meal to the first void at the morning wake-up included [22,25,26,27,28,29], an early morning blood sample collected under fasting conditions after the completion of the overnight urine collection [22,25,26,27,28,29], and a morning timed urine sample under fasting conditions collected after blood sampling [32,33]. The exam-3 protocol included the collection of an early morning blood sample under fasting conditions only [22]. Data about initiation of kidney failure replacement treatment and mortality were collected after exam-1 from the local sections of national registries. Target cohort of the present analyses were the individuals with age ≥ 18 years at exam-1 that participated also in exam-2 and exam-3.

### 2.1. Measurements

The present analyses included data collected at exam-1, exam-2, exam-3 together with mortality data from after exam-1 up to the completion of exam-3 (30 June 2007). Urinary potassium was assessed as the ratio of urinary potassium concentration to urinary creatinine concentration to exclude the bias due to errors in completeness or timing of urine collection [34] and was used as the main independent variable (from here on abbreviated as uK/Cr). Exam-1 uK/Cr in spot samples and exam-2 uK/Cr in overnight collections were separately analyzed to control whether findings were affected by circadian rhythms in urinary potassium [35,36]. Serum potassium was used as index of extracellular potassium. eGFR was calculated by the Chronic Kidney Disease—Epidemiology Collaboration equation using gender, age, and serum creatinine [31,37]. eGFR, eGFR change between exams, and incidence of decreased eGFR were used as separate, alternative, dependent variables. eGFR changes were expressed in absolute units and with data normalization per the time interval between the exams that was calculated as the difference between the exam dates.

The list of possible confounders selected for the analysis included gender, age, body mass index (weight/height^2^), estimated 24-h urinary creatinine [38], blood pressure, serum glucose, urinary sodium to creatinine ratio, urinary urea nitrogen to creatinine ratio, and data reported at questionnaires on drug treatments, smoking, habitual alcohol intake, habitual intake of water and other beverages. Urinary albumin as albumin/creatinine ratio was measured and included in the analyses only for examinees with age 45–64 years at exam-2 [39].

Blood pressure was measured by trained physicians after participants had been seated quietly for 5 min, on the right arm, with the use of mercury sphygmomanometers and cuffs of appropriate size. Three measurements were taken one minute apart and the mean of the second and third measurement were used in analyses. Serum creatinine was measured in frozen samples by automated biochemistry (Express Plus, Bayer Diagnostic) using a kinetic alkaline picrate assay with IDMS-traceable standardization [30]. Urinary potassium and other lab variables were measured in fresh samples using automated biochemistry and quality controls [21,22]. Decreased eGFR was defined as eGFR < 60 mL/min × 1.73 m^2^ [1]. Hypertension was defined as mean systolic pressure ≥ 140 mm Hg or mean diastolic pressure ≥ 90 mm Hg or regular antihypertensive drug treatment. Obesity was defined as body mass index ≥ 30 kg/m^2^. Diabetes was defined as serum glucose ≥ 7.0 mmol/L or regular anti-diabetic treatment.

### 2.2. Statistical Analyses

Descriptive data in the whole cohort were reported as prevalence for categorical variables, mean ± standard deviation (SD) for non-skewed numerical variables, and median with interquartile range (IQR) for skewed numerical variables (skewness > 1). Comparisons of skewed variables between exam-1 and exam-2 were done by Wilcoxon signed-rank test and/or by correlation analyses and paired t-test with log-transformed data. The cross-sectional relations of uK/Cr to serum potassium and to covariates were investigated using quintile analyses, separately for exam-1 and at exam-2. To reduce the effect of sex and age—that is to obtain quintiles with different uK/Cr but with similar sex and age—quintiles were defined separately for men and women in seven age-strata (18–24, 25–34, 35–44, 45–54, 55–64, 65–74 and ≥75 years). eGFR and eGFR changes were investigated along sex- and age- controlled uK/Cr quintiles without adjustments for covariates. After that, the relation of uK/Cr to eGFR or eGFR change as dependent variable was investigated in multi-variable regression models with uK/Cr and other skewed variables as log-transformed data. Five multi-variable linear models were investigated: cross-sectional relation of exam-1 uK/Cr to exam-1 eGFR (Model 1); cross-sectional relation of exam-2 uK/Cr to exam-2 eGFR (Model 2); longitudinal relation of exam-1 uK/Cr to eGFR change from exam-1 to exam-2 (Model 3); longitudinal relation of exam-2 uK/Cr to eGFR change from exam-2 to exam-3 (Model 4); longitudinal relation of the mean of uK/Cr at exam-1 and exam-2 to eGFR change from exam-1 to exam-3 (Model 5). The list of covariates in the cross-sectional Models 1–2 included gender, age, body mass index as overweight index, estimated 24-h urinary creatinine as index of urinary creatinine excretion [38], systolic pressure, diabetes, urinary sodium to creatinine ratio as index of dietary sodium intake [28,34], urinary urea nitrogen to creatinine ratio as index of dietary protein intake [25,26,27,40], smoking, habitual alcohol intake, habitual intake of water and other beverages [29], and treatment with antihypertensive drugs, diuretics, potassium-sparing diuretics, inhibitors/blockers of renin-angiotensin system. The list of covariates in the longitudinal Models 3–5 included also eGFR at initiation of the observation period that is a key predictor of eGFR changes over time in the Gubbio study cohort [25,27,28,29]. For sensitivity analyses, Model 5 was analyzed also in selected subgroups. Lastly, multi-variable logistic regression models investigated the relation of the mean of uK/Cr at exam-1 and exam-2 to after exam-2 mortality and to the incidence from exam-1 to exam-3 of decreased eGFR. For direct comparability among models, the results of linear regression were reported as standardized regression coefficient (beta) that indicates the fraction of the dependent variable standard deviation that is explained by a difference of one standard deviation of the independent variable (uK/Cr). Results included 95% confidence interval (95%CI). Statistical procedures were performed by IBM-SPSS Statistics 19.

## 3. Results

### 3.1. Descriptive Statistics

From the original cohort of the 4554 participants with age ≥ 18 years at exam-1, the analyses excluded 97 examinees with missing data at exam-1, 350 examinees who had died before exam-2, 1008 non-responders to exam-2 (exam-2 mortality-corrected response rate = 75.5%), 39 examinees with missing data at exam-2, 615 examinees who had died before exam-3, and 418 non-responders to exam-3 (exam-3 mortality-corrected response rate = 82.9%). No examinee was with missing data at exam-3. Thus, the study cohort comprised 2027 examinees (56.9% women, age at exam-1 18–74 years). Appendix A shows the flowchart for study cohort selection.

The distribution of uK/Cr was positively skewed both at exam-1 and at exam-2 (Appendix A). Daytime uK/Cr in spot sample of exam-1 was significantly higher than and positively correlated with overnight uK/Cr in timed collection of exam-2 (Appendix A). eGFR as mL/min per 1.73 m^2^ decreased from 90.5 ± 18.2 at exam-1 to 87.6 ± 14.0 at exam-2, and to 76.7 ± 14.5 at exam-3. Absolute GFR change as mL/min per 1.73 m^2^ was −2.81 ± 15.08 from exam-1 to exam-2, −10.93 ± 9.82 from exam-2 to exam-3, and −13.74 ± 15.11 from exam-1 to exam-3. Mean ± SD for time interval was 6.0 ± 1.0 years from exam-1 to exam-2, 13.3 ± 2.1 years from exam-2 to exam-3, and 19.3 ± 2.0 years from exam-1 to exam-3. With data normalization for time interval, eGFR change per year was −0.49 ± 2.64 from exam-1 to exam-2, −0.85 ± 0.76 from exam-2 to exam-3, and −0.72 ± 0.79 from exam-1 to exam-3. Prevalence of decreased eGFR was 3.2% at exam-1 (*n =* 64), 3.9% at exam-2 (*n =* 79), and 11.4% at exam-3 (*n =* 231). Incidence of decreased eGFR was 10.2% from exam-1 to exam3 (*n =* 200).

### 3.2. Analyses by Sex- and Age- Controlled uK/Cr Quintiles

At exam-1, higher uK/Cr quintile cross-sectionally associated with higher body mass index, higher diabetes prevalence, higher alcohol intake, and higher urinary sodium (Table 1).

At exam-2, higher uK/Cr quintile cross-sectionally associated with higher serum potassium, higher systolic pressure, higher urinary sodium, and higher urinary urea nitrogen (Table 2).

Regarding the relation with eGFR, higher uK/Cr quintile at exam-1 did not associate cross-sectionally with eGFR at exam-1 but it associated longitudinally with higher eGFR at exam-2 and with lesser eGFR decline from exam-1 to exam-2 (upper section of Table 3). Findings were similar without or with data normalization for time interval from exam-1 to exam-2. Higher uK/Cr quintile at exam-2 did not associate cross-sectionally with eGFR at exam-2 but it associated longitudinally with higher exam-3 eGFR and with lesser eGFR decline from exam-2 to exam-3 (central section of Table 3). Findings were similar without or with data normalization for time interval from exam-2 to exam-3. Higher quintile of the uK/Cr mean between exam-1 and exam-2 associated longitudinally with higher exam-3 eGFR and with lesser eGFR decline from exam-1 to exam-3 (lower section of Table 3). Findings were similar without or with data normalization for time interval from exam-1 to exam-3.

### 3.3. Multivariable Regression Analyses

Log-transformed uK/Cr did not independently relate to eGFR of the same exam either at exam-1 and at exam-2 (Table 4, Models 1 and 2). Log-transformed uK/Cr, both at exam-1 and at exam-2, positively and independently related to eGFR change at the subsequent exam (Table 4, Models 3–4). Mean of log transformed uK/Cr at exam-1 and exam-2 positively and independently related to eGFR change from exam-1 to exam-3 (Table 4, Model 5).

Findings of Model 5 were similar in sensitivity analyses for the following groups: men and women, age 18–44, 45–64 and 65 and over, eGFR < 90 and eGFR ≥ 90 mL/min per 1.73 m^2^, hypertensive and non-hypertensive, obese and non-obese, diabetic and non-diabetic, drinker and non-drinker, and with urinary sodium or urinary urea nitrogen below or above the median (Figure 1).

Findings of multivariable Model 5 were similar when the analysis was repeated with control also for urinary albumin/creatinine ratio in the 943 examinees with age 45–64 at exam-2 and measured urinary albumin/creatinine ratio (beta = 0.046, 95%CI = 0.01/0.091, *p* = 0.044; median and IQR of urinary albumin/creatinine ratio = 3.40 and 1.08/6.04 mg/g).

In the logistic multivariable model with control for the same covariates included in Model 5 (footer of Table 4), one SD higher mean of log-transformed uK/Cr at exam-1 and exam-2 (+0.16 log mmol/g) did not relate to after exam-2 mortality (odds ratio = 0.99, 95%CI = 0.85/1.16, *p* = 0.889) but it related to 22% lower incidence of decreased eGFR from exam-1 to exam-3 (odds ratio = 0.78, 95%CI = 0.61/0.98, *p* = 0.039).

## 4. Discussion

The results of the present long-term observational study indicated that, in a sample of the adult general Italian population, higher uK/Cr ratio did not relate cross-sectionally to eGFR but related to a lesser eGFR decline during an observation period ranging approximately from 6 to 20 years, independently of gender, age, and other variables.

The main limitation of the study was the lack of 24-h urine collection which is considered the best marker of dietary potassium intake. Thus, the possibility cannot be excluded that the findings reflected the influence of circadian rhythms of urinary potassium rather than the influence of dietary potassium intake. If this were the case, the study results would indicate that the circadian rhythm in urinary potassium is an independent predictor of kidney function decline in the population. However, two observations were against a possible key role of circadian rhythms. First, the relation of uK/Cr with eGFR change over time was substantially identical in analyses for daytime uK/Cr at exam-1 and for overnight uK/Cr at exam-2. Second, quintiles with higher overnight uK/Cr should have lower morning serum potassium at the end of the overnight urine collection if higher overnight uK/Cr was not sustained by a higher potassium intake in the previous meal and was caused only by an increased excretion due to circadian rhythms. Vice versa, exam-2 data in Table 2 indicated that quintiles with higher overnight uK/Cr had higher morning serum potassium, a finding that could be explained only by parallel influences of a higher dietary potassium intake on urine potassium and serum potassium. In accordance with the interpretation that higher uK/Cr reflected higher dietary K intake there were the positive associations of uK/Cr with other diet-related indices both at exam-1 (body mass index, alcohol intake, and urinary sodium) and at exam-2 (urinary sodium and urinary urea nitrogen). Due to the day-to-day intra-individual variability in potassium intake and urinary potassium [41,42], a single measurement of urinary potassium was expected to imply a regression dilution bias and an underestimate of the true strength of the association. The observation that the highest beta was found with the use of the mean of exam-1 uK/Cr and of exam-2 uK/Cr was in accordance with this expectation. Another limitation was the lack of data on serum cystatin C that could have improved the accuracy of eGFR [43].

An observational, population-based study can hardly clarify the mechanism(s) underlying the observed associations. At present, the mechanisms remain hypothetical for the relation of higher urinary potassium with lesser eGFR decline over time. A direct favorable effect of potassium intake could be involved if high potassium intake would play a direct protective role against kidney damage also in humans, as described in experimental models [44,45]. If potassium had direct favorable effects, a reno-protective role could be conceivable also for drugs affecting potassium homeostasis [5,6]. The effects of high potassium intake on kidney function decline could be mediated also by indirect, sodium-dependent mechanism(s). As recently reviewed by Wey et al. [12], high potassium intake stimulates natriuresis reducing the activity of the thiazide-sensitive sodium-chloride cotransporter in the kidney distal convoluted tubule. Given that high sodium intake related to greater kidney function decline in clinical studies and in population-based studies [28,46], the natriuretic effects of high potassium intake could be an indirect pathway linking potassium intake to kidney function decline. Other possibilities cannot be excluded.

Urinary potassium did not relate to eGFR decline in the seminal paper of Kieneker et al. when analyses were controlled also for baseline eGFR and albuminuria [7], nor in the paper of Deriaz et al. when analyses were done for urinary potassium by itself [8]. Therefore, this is the first population-based report of an independent relation of higher urinary potassium with lesser eGFR decline over time. Regarding previous clinical studies on urinary potassium, the present results agreed with the observations on Japanese diabetics [14,15], on Dutch outpatients with eGFR ≥ 60 mL/min per 1.73 m^2^ [16], on Korean non-dialysis patients with chronic kidney disease stages 1–5 [17], and on high cardiovascular risk patients of multicenter trials [18] but disagreed with observations on US patients of the Modification of Diet in Renal Disease study and of the Chronic Renal Insufficiency Cohort Study [19,20]. Given that potassium content is high not only in fruits like banana but also in meat [47], the disagreement among studies in different countries could be due also to the predominant sources of habitual dietary potassium in the specific country under study. According to this possibility, the relation of dietary potassium with kidney function decline could differ between populations with high potassium intake due to a steak-rich diet and population-samples with high potassium intake due to veg-rich diet.

Regarding practical implications, the study results support the suggestion of high potassium diet in individuals at risk of kidney function decline on the basis of the significant and independent relation of higher urinary potassium with lesser eGFR decline. Although the relation was consistent in this study also in the subgroup with eGFR < 90 mL/min × 1.73 m^2^, two points recommend a word of caution for the expansion of the suggestion in chronic kidney disease stage 3 or higher: the present observation of higher serum potassium in people with higher urinary potassium and the previous report of higher mortality in people with serum potassium higher than 5 mmol/L [48].

Briefly, this observational cohort study reported that, in a sample of the adult general population, a higher urinary potassium related to a lesser eGFR decline during an observation period approximately ranging from 6 to 20 years, and independently of gender, age, and several other variables. Altogether, the results supported the hypothesis that a high dietary potassium intake could have favorable effects against the progressive decline in kidney function associated with ageing. Moreover, study results underlined the need of investigations about the relation of dietary potassium on kidney function with focus also on the source of dietary potassium.

## Figures and Tables

**Figure 1 nutrients-13-02747-f001:**
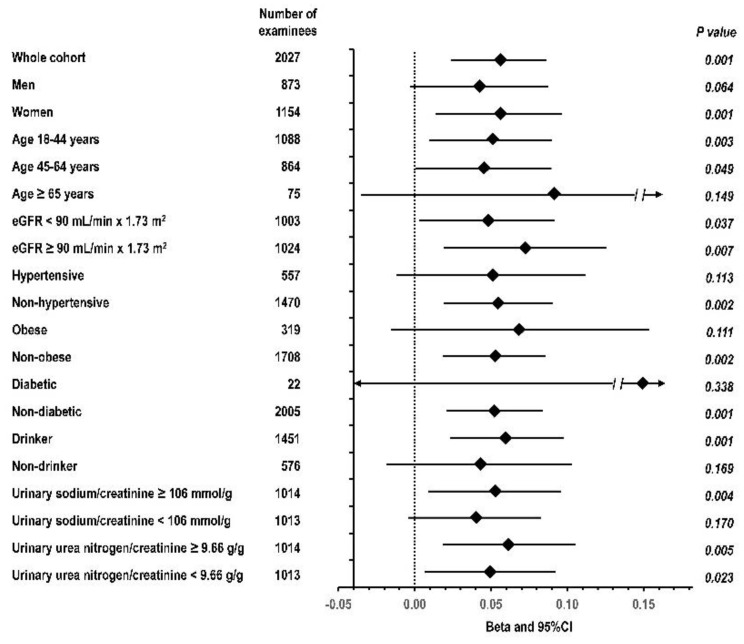
Multivariable standardized regression coefficient (beta) with 95% confidence interval and P value for the relation of the mean of uK/Cr at exam-1 and exam-2 with yearly eGFR change from exam-1 to exam-3: whole study cohort and selected subgroups. The dotted line indicates beta = 0. Covariates included in the multivariable model are listed in the footer of Table 4. Subgroups were defined on the basis of data collected at exam-1.

**Table 1 nutrients-13-02747-t001:** Descriptive statistics for data of exam-1 in whole cohort (prevalence for categorical variables, mean ± SD for non-skewed numerical variables, and median with interquartile range for skewed variables) and trend analyses by quintiles of uK/Cr (prevalence or mean).

	Whole Cohort	Quintile * of uK/Cr at Exam-1					*p* for Trend ^¶^
		1	2	3	4	5	
N	2027	402	439	377	406	403	
Women, %	56.9%	57.2%	52.6%	61.5%	56.9%	57.1%	0.576
Age, years	42.9 ± 13.3	42.6	42.0	44.4	42.8	43.0	0.439
uK/Cr ^†^, mmol/g	28.6 (21.7/38.6)	17.3	23.3	29.4	36.2	60.3	<0.001
Serum potassium, mmol/L	4.04 ± 0.36	4.05	4.03	4.03	4.05	4.02	0.589
Body mass index, kg/m^3^	26.1 ± 4.2	26.2	25.6	26.2	25.9	26.8	0.015
Estimated urinary creatinine, g/d	1.26 ± 0.31	1.27	1.27	1.23	1.26	1.28	0.993
Systolic pressure, mmHg	127 ± 18	127	127	128	126	127	0.378
Diastolic pressure, mmHg	76 ± 11	77	76	77	75	76	0.287
Antihypertensive drug treatment, %	9.0%	7.7%	9.8%	9.3%	9.1%	8.9%	0.710
Diuretic, %	7.7%	6.7%	8,7%	8.8%	6.9%	7.7%	0.981
Potassium sparing diuretic, %	2.2%	1.7%	1.8%	2.4%	2.2%	3.0%	0.213
Inhibitor/blocker renin-angiotensin system, %	0.4%	0.2%	0.7%	0.3%	0.7%	0.0%	0.639
Diabetes, %	1.1%	1.0%	0.2%	1.1%	1.0%	2.2%	0.044
Smoking, %	15.5%	16.2%	13.4%	16.2%	16.5%	15.4%	0.768
Habitual alcohol intake, g/d	11.9 (0.0/35.7)	16.4	20.8	19.4	22.1	23.2	0.001
Urine sodium/creatinine ^†^, mmol/g	106 (74/151)	96	108	119	123	162	<0.001
Urine urea nitrogen/creatinine, mmol/g	Not measured	Not measured					---

* defined separately by gender and age stratum (years: 18–24, 25–34, 35–44, 45–54, 55–64, ≥65); ^¶^ by chi-square analysis or ANOVA; ^†^ daytime spot sample.

**Table 2 nutrients-13-02747-t002:** Descriptive statistics for data of exam-2 in whole cohort (prevalence for categorical variables, mean ± SD for non-skewed numerical variables, and median with interquartile range for skewed variables) and trend analyses by quintiles of uK/Cr (prevalence or mean).

	Whole Cohort	Quintile * of uK/Cr at Exam-2					*p* for Trend ^¶^
		1	2	3	4	5	
N	2027	398	399	401	401	428	
Women, %	56.9%	56.5%	56.4%	55.6%	56.1%	59.8%	0.400
Age, years	48.9 ± 13.3	48.4	48.6	49.2	49.0	49.0	0.467
uK/Cr ^†^, mmol/g	24.3 (18.7/32.9)	14.8	20.2	24.6	30.2	47.0	<0.001
Serum potassium, mmol/L	4.12 ± 0.34	4.10	4.10	4.12	4.11	4.17	0.005
Body mass index, kg/m^3^	26.9 ± 4.3	26.8	26.7	26.7	26.9	27.2	0.149
Estimated urinary creatinine, g/d	1.25 ± 0.31	1.26	1.25	1.25	1.26	1.24	0.652
Systolic pressure, mmHg	125 ± 18	123	125	126	126	126	0.016
Diastolic pressure, mmHg	76 ± 10	75	75	76	76	75	0.359
Antihypertensive drug treatment, %	14.9%	12.3%	15.0%	13.7%	16.7%	16.8%	0.056
Diuretic, %	7.5%	4.5%	9.8%	6.7%	7.0%	9.3%	0.093
Potassium sparing diuretic, %	3.4%	3.3%	2.8%	4.0%	4.0%	3.0%	0.804
Inhibitor/blocker renin-angiotensin system, %	4.2%	2.5%	4.0%	4.5%	5.7%	4.4%	0.083
Diabetes, %	3.3%	2.8%	3.5%	4.7%	2.5%	3.0%	0.858
Smoking, %	31.6%	32.7%	34.1%	34.2%	28.9%	28.3%	0.053
Habitual alcohol intake, g/d	11.9 (0.0/35.7)	20.8	22.2	24.2	22.9	22.4	0.400
Urine sodium/creatinine ^†^, mmol/g	103 (68/146)	82	101	113	122	156	<0.001
Urine urea nitrogen/creatinine ^†^, mmol/g	9.66 (7.50/12.26)	8.71	9.62	10.56	10.66	12.64	<0.001

* defined separately by gender and age stratum (years: 18–24, 25–34, 35–44, 45–54, 55–64, ≥65); ^¶^ by chi-square analysis or ANOVA; ^†^ overnight timed collection.

**Table 3 nutrients-13-02747-t003:** eGFR and eGFR changes by quintile of uK/Cr (prevalence or mean).

	Quintile * of uK/Cr at Exam-1					*p* for Trend ^¶^
	1	2	3	4	5	
Number of examinees	402	439	377	406	403
Time from exam 1 to exam 2, years	5.96	5.90	5.97	5.89	6.03	0.372
eGFR, mL/min × 1.73 m^2^						
at exam-1	90.6	91.8	87.8	90.4	91.3	0.990
at exam-2	86.5	88.0	85.1	88.6	88.8	0.023
change from exam-1 to exam-2	−4.06	−3.75	−1.78	−1.79	−2.54	0.033
yearly change from exam-1 to exam-2	−0.737	−0.671	−0.248	−0.314	−0.460	0.026
	**Quintile * of uK/Cr at Exam-2**					
	**1**	**2**	**3**	**4**	**5**	
Number of examinees	398	399	401	401	428	
Time from exam 2 to exam 3, years	13.5	13.5	13.3	13.3	13.0	0.001
eGFR, mL/min × 1.73 m^2^						
at exam-2	86.5	87.6	88.4	87.6	88.0	0.171
at exam-3	75.0	76.4	76.7	77.5	78.0	0.002
change from exam-2 to exam-3	−11.50	−11.24	−11.66	−10.16	−10.04	0.009
yearly change from exam-2 to exam-3	−0.887	−0.870	−0.908	−0.788	−0.799	0.032
	**Quintile * of Mean uK/Cr at Exam−1 and Exam−2**					
	**1**	**2**	**3**	**4**	**5**	
Number of examinees	395	447	378	405	402	
Time from exam-1 to exam-3, years	19.3	19.1	19.3	19.3	19.4	0.241
eGFR, mL/min × 1.73 m^2^						
at exam-1	90.2	89.1	90.8	91.7	90.3	0.363
at exam-3	75.3	75.3	77.4	77.7	77.6	0.003
change from exam-1 to exam-3	−14.89	−13.88	−13.34	−13.20	−12.69	0.034
yearly change from exam-1 to exam-3	−0.788	−0.733	−0.701	−0.683	−0.666	0.020

* defined separately by gender and age stratum (years: 18–24, 25–34, 35–44, 45–54, 55–64, ≥65). ^¶^ by ANOVA.

**Table 4 nutrients-13-02747-t004:** Standardized regression coefficient of log transformed uK/Cr to eGFR at the same exam and to eGFR change over time at subsequent exams (beta and 95%CI).

Type of Model	Independent Variable	Model	Dependent Variable	Beta (95%CI)
Cross-sectional	log transformed uK/Cr at exam-1	1 ^a^	eGFR at exam-1	−0.020 ^ns^ (−0.059/0.019)
	log transformed uK/Cr at exam-2	2 ^b^	eGFR at exam-2	0.024 ^ns^ (−0.013/0.056)
Longitudinal	log transformed uK/Cr at exam-1	3 ^c^	Yearly eGFR change from exam-1 to exam-2	0.051 ** (0.018/0.084)
	log transformed uK/Cr at exam-2	4 ^d^	Yearly eGFR change from exam-2 to exam-3	0.048 * (0.005/0.091)
	mean of log transformed uK/Cr at exam-1 and exam-2	5 ^e^	Yearly eGFR change from exam-1 to exam-3	0.056 *** (0.027/0.087)

^ns^ not significant (*p >* 0.05), * *p <* 0.03, ** *p <* 0.01, *** *p* ≤ 0.001. ^a^ covariates in the model: gender and exam-1 data for age, estimated 24-h urinary creatinine, body mass index, systolic pressure, log-transformed urinary sodium/creatinine ratio, log alcohol intake, and categorical values for diabetes, treatment with antihypertensive drugs, diuretics, potassium-sparing diuretics, inhibitors/blockers of renin-angiotensin system (yes/no = 1/0). ^b^ covariates in the model: gender and exam-2 data for age, estimated 24-h urinary creatinine, body mass index, systolic pressure, log-transformed urinary sodium/creatinine ratio, log alcohol intake, log urinary urea nitrogen/creatinine ratio, and categorical values for diabetes, treatment with antihypertensive drugs, diuretics, potassium-sparing diuretics, inhibitors/blockers of renin-angiotensin system (yes/no = 1/0). ^c^ covariates in the model: same as in Model 1 with addition of exam-1 eGFR. ^d^ covariates in the model: same as in Model 2 with addition of exam-2 eGFR. ^e^ covariates in the model: gender, exam-1 age, exam-1 eGFR, mean of exam-1 and exam-2 data for estimated 24-h urinary creatinine, body mass index, systolic pressure, log alcohol intake, log urinary sodium/creatinine ratio, exam-2 log urinary urea nitrogen/creatinine ratio, and categorical values for the prevalence at exam-1 and the incidence at exam-2 of diabetes, treatment with antihypertensive drugs, diuretics, potassium-sparing diuretics, inhibitors/blockers of renin-angiotensin system (yes/no = 1/0).

## Data Availability

M.C. and M.L. had access to the data at the Centro Studi Epidemiologici di Gubbio.

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
