# Peer review of "Urinary Potassium and Kidney Function Decline in the Population—Observational Study"

_nutrients, 2021, doi:10.3390/nu13082747_

Round 1
Reviewer 1 Report
I congratulate the Authors on the idea, implementation, and form of presenting the research results in the manuscript. At the same time, please pay attention to the suggestions:
- Materials and Methods
The serum potassium concentration cannot be used as an index of potassium homeostasis as it is mainly an intracellular ion.
2. Discussion
Discussion is well-written and can help with the interpretation results of the research by medical practitioners.
Author Response
The authors do thank the reviewer for the nice comments.
Reviewer’s comment: The serum potassium concentration cannot be used as an index of potassium homeostasis as it is mainly an intracellular ion.
Authors’ response: The authors agree that serum potassium is not an adequate index of serum homeostasis. The specific sentence in Methods was modified accordingly (line 122 of the revised version). The new text is highlighted with red character.
Reviewer 2 Report
The Present manuscript describes the correlation between urinary potassium and kidney functions. The manuscript is well written and a good manuscript with great clinical significance. I suggest the authors improve the manuscript based on the following suggestions.
Introduction:
- Add few sentences about the role of K+ voltage-gated channel and Na+K+ ATPase in the ion homeostasis
- SGLT-2 and its inhibitor (empagliflozin) in renal protection
- Antihypertensive drugs and renal outcome
- RAS inhibitors in the renal outcome:
Briefly describe their mechanisms
- Methodology for measuring eGFR
Results:
- Albumin-to-creatinine ratio in the cohort.
- Changes in Cystatin C levels
Discussion:
- Describe new renal protective approaches and cell signaling molecules such as glucocorticoid receptors, FGF receptors, or available medications and mechanisms that are linked to kidney K+ homeostasis.
- A schematic chart or diagram concluding this hypothesis
Author Response
The authors do thank the reviewer for nice comments and helpful suggestions. All points raised by the reviewer were addressed in the revised version as detailed below.
Introduction
Add few sentences about the role of K+ voltage-gated channel and Na+K+ ATPase in the ion homeostasis
Authors’ response: Four sentences were added in the revised version (lines 46-57). The new text is highlighted with red character.
SGLT-2 and its inhibitor (empagliflozin) in renal protection
Antihypertensive drugs and renal outcome
RAS inhibitors in the renal outcome:
Briefly describe their mechanisms
Authors’ response: Four sentences were added in the revised version (lines 37-44). The new text is highlighted with red character.
Methodology for measuring eGFR
Authors’ response: Three sentences were added in the revised version (lines 80-86). The new text is highlighted with red character.
Results
Albumin-to-creatinine ratio in the cohort.
Authors’ response: Authors’ response: Median and interquartile range of urinary albumin/creatinine ratio was added in Results of the revised version together with data of the specific multi-variable model 5 (line 274 of the revised version). The new text is highlighted with red character.
Changes in Cystatin C levels
Authors’ response: Serum cystatin C was not measured in the Gubbio study. This point was added to the study limitations (lines 308-309). The new text is highlighted with red character.
Discussion
Reviewer’s comment: Describe new renal protective approaches and cell signaling molecules such as glucocorticoid receptors, FGF receptors, or available medications and mechanisms that are linked to kidney K+ homeostasis. A schematic chart or diagram concluding this hypothesis.
Authors’ response: The authors’ opinion is that an epidemiological observational study is a weak method to clarify the mechanism(s) underlying an association (lines 310-311). This is why the discussion on mechanisms was limited to few sentences. For the same reason the authors preferred not to include a hypothetic diagram. A sentence was added about the possible role of drugs affecting potassium homeostasis (lines 315-316). The new text is highlighted with red character.
The reference list was modified accordingly to the changes in the text and seven new quotations were added (#1-6 and 43). New references are highlighted with red character.
Round 2
Reviewer 2 Report
The authors have revised and improved the manuscripts.